# Cultivation of *Prevotella copri* in a medium supplemented with tomato juice suppresses the bacteria-induced intestinal permeability in *Caenorhabditis elegans*

Nobuo Fuke[1]*, Natsumi Desaka[2], Yuichiro Nakazawa[1], Shigenori Suzuki[1], Kenji Matsumoto[2], Yasuki Higashimura[2]*

**1** Diet & Well-being Research Institute, KAGOME CO., LTD., Nasushiobara, Tochigi, Japan, **2** Department of Food Science, Ishikawa Prefectural University, Nonoichi, Ishikawa, Japan

* nobuo_fuke@kagome.co.jp (NF); yasuki@ishikawa-pu.ac.jp (YH)

## Abstract

Epidemiological studies in humans have suggested that tomato consumption and the compositional ratios of *Prevotella*, *Megamonas,* and *Streptococcus* in the intestinal microbiota are related to intestinal permeability. In this study, we investigated the causal relationship using *Caenorhabditis* (*C.*) *elegans*. We cultured *Escherichia coli* OP50 (the standard *C. elegans* food), *Prevotella* (*P.*) *copri* JCM 13464[T], *Megamonas funiformis* JCM 14723[T], and *Streptococcus salivarius* JCM 5707[T] in either normal medium or medium containing 1% (v/v) tomato juice (TJ medium), fed these bacteria to *C. elegans* for three days, and evaluated intestinal permeability using the Smurf assay. The proportion of Smurf individuals was significantly lower in *C. elegans* fed *P. copri* cultured in TJ medium than in those fed the same bacteria cultured in normal medium, while other strains showed no such medium-dependent differences. Interestingly, heat-killed *P. copri* grown in normal medium also reduced the proportion of Smurf individuals. Furthermore, *P. copri* grown in TJ medium exhibited a lower survival rate after seeding on nematode growth medium, an effect not observed in other strains. Liquid chromatography-tandem mass spectrometry analysis revealed that *P. copri* cultured in TJ medium accumulated L-(-)-3-phenyllactic acid (L-(-)-3-PLA), a compound known for its antibacterial properties through oxidative stress and its protective effects on the intestinal barrier. In contrast, the levels of known antioxidants such as 2,3,4,9-tetrahydro-1H-β-carboline-3-carboxylic acid and Cyclo(phenylalanyl-prolyl) were decreased. Culturing *P. copri* in normal medium supplemented with L-(-)-3-PLA alone did not reduce survival, suggesting that both L-(-)-3-PLA accumulation and the depletion of antioxidants contribute to reduced viability. Additionally, L-(-)-3-PLA directly suppressed intestinal permeability in *C. elegans*. In conclusion, the results of this study suggest that TJ may inhibit increased intestinal permeability through both rendering *P. copri* vulnerable and the direct effects of L-(-)-3-PLA. Further studies are needed to determine the relevance of these findings to humans.

**Data availability statement:** All relevant data are within the paper and its Supporting information files.

**Funding:** This study was self-funded by KAGOME CO., LTD. (https://www.kagome.co.jp/). The authors N.F., Y.N., and S.S. are employees of KAGOME CO., LTD. The sponsor was involved in the study design, data collection and analysis, decision to publish, and preparation of the manuscript.

**Competing interests:** I have read the journal's policy and the authors of this manuscript have the following competing interests: KAGOME CO., LTD. produces and sells tomato juice. The authors N.F., Y.N., and S.S. hold stock in KAGOME CO., LTD. This study was conducted as a collaborative research between KAGOME CO., LTD. and Ishikawa Prefectural University. This does not alter our adherence to the PLOS ONE policy on sharing data and materials.

## Introduction

Disruption of the intestinal barrier and increased intestinal permeability have been implicated in a range of pathological conditions. In particular, it has been suggested that the influx of lipopolysaccharide (LPS), an outer membrane component of Gram-negative bacteria, into the bloodstream causes chronic inflammation and contributes to the development and progression of many lifestyle-related diseases including obesity [1], insulin resistance [2], diabetes [3], and nonalcoholic fatty liver disease [4]. Accordingly, identifying dietary components that mitigate intestinal permeability and elucidating their underlying mechanisms is a critical objective in the field of preventive medicine.

In a large epidemiological study, we previously found that tomato intake was negatively correlated with blood LPS binding protein (LBP) concentration, which is an alternative indicator of LPS exposure [5]. In that study, we found that three genera, *Prevotella*, *Megamonas*, and *Streptococcus*, were positively correlated with LBP levels in blood. Among these genera, tomato intake was negatively correlated with the composition of *Streptococcus*. These findings suggest that tomato intake may reduce intestinal permeability by modulating the gut microbiota. However, as these observations are based on cross-sectional data, the causal relationships remain to be clarified.

The nematode *Caenorhabditis elegans* (*C. elegans*) are free-living organisms that are approximately 1 mm long and primarily feed on bacteria. Because the nematode body is transparent and its intestinal tract can be easily observed, it has attracted attention as a model for studying the interaction between intestinal bacteria and the intestinal tract, which were previously difficult to observe [6,7]. In addition, because approximately half of the genes in *C. elegans* are orthologs of human genes and share similar metabolic pathways, *C. elegans* is used as animal models for humans [8]. Among the tight junction-related proteins involved in the human intestinal barrier, claudin-like proteins have been identified in *C. elegans*, and their knockdown has been reported to increase intestinal permeability [9]. The Smurf assay was established to assess intestinal permeability by inducing nematodes to ingest dyes that normally do not pass through the intestinal tract, using dye leakage as an indicator [10].

In this study, we evaluated the effects of tomato consumption on *Prevotella*, *Megamonas*, and *Streptococcus* and the intestinal permeability induced by these bacteria using *C. elegans*. *C. elegans* are a bacterivore and do not consume tomatoes directly. Therefore, we cultured each bacterium in a standard medium for intestinal bacteria (BG medium) and BG medium supplemented with 1% (v/v) tomato juice (TJ medium), fed them to *C. elegans*, and evaluated intestinal permeability using the Smurf assay. Commercially available tomato juice was selected as the test sample due to its consistent quality compared to fresh juice and its suitability for use in future human intervention studies. Additionally, based on our previous findings, we confirmed that adding food ingredients to the gut microbiota culture system at a concentration of 1% (v/v) is sufficient to alter the composition of various bacterial species. Furthermore, at a concentration of 1% (v/v), we believe the impact on bacteria can be

assessed without disrupting the nutrient composition inherent to the BG medium. Thus, a 1% (v/v) concentration of tomato juice was chosen for this study. As previously published reports have not specified the species of *Prevotella*, *Megamonas*, and *Streptococcus* associated with blood LBP concentrations, we selected species from these genera that have been suggested to be associated with blood LPS and used their reference strains: *Prevotella* (*P.*) *copri* JCM 13464[T], *Megamonas* (*M.*) *funiformis JCM* 14723[T], and *Streptococcus* (*S.*) *salivarius JCM* 5707[T]. To determine the mechanism of action of TJ, the medium, culture supernatant, and cultured bacteria were analyzed using liquid chromatography-tandem mass spectrometry (LC-MS/MS). Among these three bacteria, only *P. copri* JCM 13464[T] showed an inhibitory effect on intestinal permeability when cultured in the TJ medium. Culturing on TJ medium reduced the survival rate of *P. copri* JCM 13464[T] after application to nematode growth medium (NGM), and L-(-)-3-phenyllactic acid (L-(-)-3-PLA) was identified as a potential contributingfactor.

## Materials and methods

### *C. elegans* strain

The *C. elegans* Bristol N2 strain was obtained from Caenorhabditis Genetics Center (Minneapolis, MN) and maintained at 20°C on NGM seeded with *Escherichia* (*E.*) *coli* OP50 (Caenorhabditis Genetics Center).

### Bacterial strains

The bacterial strains used in this study are summarized in Table 1. All strains, except for *E. coli* OP50, were obtained from the Microbe Division/Japan Collection of Microorganisms (JCM), RIKEN BioResource Research Center (Ibaraki, Japan).

### Preparation of NGM plates with test bacteria

Bacteria were cultured anaerobically using AnaeroPack (Mitsubishi Gas Chemical Company, Inc., Tokyo, Japan) in BG medium, a 7:3 mixture of Bryant and Burkey medium (Biokar Diagnostics-Solabia Group, Pantin, France) and GAM medium (Shimadzu Diagnostics Corporation, Tokyo, Japan), which has been validated as a screening platform for evaluating microbiome-derived drug metabolism while maintaining donor-specific microbial communities, including anaerobes [14]. Specifically, 100 mL of BG medium, BG medium supplemented with 1% (v/v) commercial 100% tomato juice (KAGOME CO., LTD., no salt added, sterilized at 121°C for 15 minutes) (TJ medium), or BG medium supplemented with 5.5 μM of L-(-)-3-PLA was inoculated with 1% (v/v) of each bacterial culture and incubated overnight at 37°C. After incubation, cultures were filtered through a 40 μm-diameter strainer (AS ONE Corporation, Osaka, Japan) to remove tomato-derived solids, centrifuged at 8,000×g for 10 minutes at 4°C, and the supernatant was collected. The bacterial pellet was resuspended in 40 mL of culture supernatant. A portion of the culture medium was then aliquoted, and its absorbance at 660 nm was measured using a Spectrophotometer U-2910 (Hitachi High-Tech Science Corporation, Tokyo, Japan). The culture medium was further centrifuged at 8,000×g for 10 minutes at 4°C, and the culture supernatant was collected. The remaining pellet was resuspended in an appropriate volume of culture supernatant, such that the final absorbance at 660 nm was equivalent to 100. This concentrated bacterial solution (50 μL) was applied to the center of the NGM and was used as the test bacteria-seeded NGM.

**Table 1. Bacterial strains used in this study.**

| Species | Strain | Selection criteria |
| --- | --- | --- |
| *Prevotella copri* | JCM 13464[T] | *Prevotella* species reported to correlate with serum LPS concentrations [11,12]. |
| *Megamonas funiformis* | JCM 14723[T] | The sole commercially available *Megamonas* strain of human fecal origin. |
| *Streptococcus salivarius* | JCM 5707[T] | *Streptococcus* species reported to correlate with serum LPS concentrations [13]. |

To examine the localization of components involved in intestinal permeability in *P. copri* JCM 13464^T, the bacteria were cultured as described above to prepare a bacterial concentrate, a portion of which was then centrifuged at 8,000 × g for 10 minutes at 4°C, and the pellet was suspended in 40 mL of sterile distilled water. After centrifugation and two washes, the pellet was resuspended in 40 mL of each culture supernatant. After further centrifugation at 8,000 × g for 10 minutes at 4°C, the remaining pellet was resuspended in an appropriate volume of culture supernatant so that the final absorbance at 660 nm was equivalent to 100. The pellets were resuspended in either their original culture supernatant or the supernatant from other culture conditions.

In the ingestion study of heat-killed *P. copri* JCM 13464^T, the bacterial solution, cultured and concentrated as described above, was subjected to heat treatment at 60°C for 30 minutes; the conditions used to inactivate *Prevotella* were previously reported [15]. The heated bacteria were allowed to cool to room temperature and then applied to NGM.

NGM containing L-(-)-3-PLA was prepared according to previously reported methods [16]. Specifically, L-(-)-3-PLA was added to the NGM to achieve final concentrations of 10 nM or 100 nM during the cooling phase, after autoclaving and before solidification. After the NGM had cooled and solidified, 50 µL of the cultured and concentrated *E. coli* OP50 suspension, as described above, was applied to the agar.

## Cultivation *of C. elegans*

To evaluate the potential effects of the test bacterial strains on *C. elegans* development, we examined whether administering these strains from the time of hatching would influence worm growth. In control animals fed *E. coli* OP50 cultured on BG medium, body size increased steadily through day 5 post-hatching (S1 Table). In contrast, worms fed *P. copri* JCM 13464^T, *M. funiformis* JCM 14723^T, or *S. salivarius* JCM 5707^T from hatching exhibited slower body growth. Furthermore, worms fed *P. copri* JCM 13464^T exhibited a marked reduction in egg production (S2 Table). Based on these findings, to minimize any confounding effects of the test strains on somatic development, we began administration of the bacteria on day 5 post-hatching, when body growth is considered to plateau. The final *C. elegans* feeding schedule was as follows: Three milliliters of S-basal buffer (0.584% [w/v] sodium chloride, 50 mM potassium phosphate, pH 6.0) was added to the NGM, in which *C. elegans* was passively reared, and *C. elegans* was suspended on the surface of the medium by gentle shaking. This was collected in a 15 mL tube and allowed to stand for 5 minutes to allow *C. elegans* to settle to the bottom of the tube. The supernatant was removed using a pipette, and 5 mL of S-basal buffer, 500 µL of chlorine bleach (Kao Corporation, Tokyo, Japan), and 100 µL of 10N sodium hydroxide solution were added. *C. elegans* was lysed by vigorous manual shaking for 5 minutes, followed by centrifugation at 400 × g for 30 s at room temperature. The supernatant was discarded, and 13 mL S-basal buffer was added, and the mixture was stirred using a vortex mixer. The centrifugation and washing procedures were repeated six times. After washing, the eggs were suspended in 2 mL of S-basal buffer and incubated at 20°C in inverted tubes. The next day, the number of hatched *C. elegans* was counted, and a suspension containing 750 eggs was dispensed onto the *E. coli* OP50-seeded NGM. The medium was incubated at 20°C for 3 days. To prevent progeny contamination, 5-fluoro-2'-deoxyuridine (FUdR; FUJIFILM Corporation, Tokyo, Japan) was added dropwise to the NGM plates containing *C. elegans* at a final concentration of 0.5 mg/mL on day 3 (day 4 post-hatching). On the following day (day 5 after hatching), S-basal buffer was added to the NGM, and *C. elegans* were suspended by shaking and collected in tubes. The *C. elegans* suspension was allowed to stand for 5 minutes to let the worms settle at the bottom of the tube, after which the supernatant was removed. Next, 14 mL of fresh S-basal buffer was added and *E. coli* OP50 on the surface of *C. elegans* was removed by gentle inversion. The washing procedure was repeated five times in total. In addition, *C. elegans* were transferred onto plain NGM and incubated at 20°C for 1 hour to expel any remaining *E. coli* OP50 in the intestinal tract. Three milliliters of S-basal buffer were added to the NGM, and *C. elegans* were suspended by shaking and collected in tubes. As previously mentioned, *C. elegans* were washed twice with S-basal buffer, the supernatant was removed, and the worms were suspended in 2 mL of S-basal buffer. Preliminary experiments confirmed the absence of viable *E. coli* OP50 cells following this washing procedure. *C. elegans* were then transferred to the

test bacteria-seeded NGM plates with drops of FUdR at approximately 100 animals per group and incubated at 20°C until day 8 post-hatching. On days 6 and 7 after hatching, *C. elegans* were transferred to a new test bacteria-seeded NGM with FUdR drops to minimize the possibility of contamination with *E. coli* OP50. On day 8 after hatching, the *C. elegans* were subjected to the Smurf assay.

### Smurf assay

*C. elegans* were transferred to a 1:1 mixture of 10% (w/v) Acid Blue (Tokyo Chemical Industry Co., Ltd., Tokyo, Japan) and S-basal buffer (400 µL each), and incubated at 20°C for 3 hours. During incubation, the mixture was constantly mixed using a rotator. After a gentle spin-down to dislodge *C. elegans* from the cap, the tube was left standing for 3 minutes to allow the *C. elegans* to settle. After removing the supernatant, 1.5 mL of S-basal buffer was added to wash *C. elegans*. The tubes were left to stand for 1 minute to allow the *C. elegans* to settle; the supernatant was removed, and S-basal buffer was added again. The washing procedure was repeated until the supernatant was clear. The supernatant was removed as thoroughly as possible, and 30 µL of 1 M sodium azide solution was added to immobilize *C. elegans*. The supernatant was as completely as possible, and the remaining *C. elegans* suspension was transferred to a glass slide. After being covered with a cover glass, the samples were observed under a microscope. An optical microscope (Olympus BX51; Olympus Corporation, Tokyo, Japan) was used for observation, with 4 × magnification by an objective lens and 45× with a CCD camera and monitor. Worms exhibiting leakage of blue dye from the intestinal tract were classified as having increased intestinal permeability ("Smurf" phenotype).

### Observation with scanning electron microscopy

The *P. copri* JCM 13464[T] bacterial concentrate was divided into three tubes, and all samples were centrifuged at 8,000 × g for 10 minutes at 4°C. Sterile distilled water (40 mL) was added to the pellet in two tubes to suspend the bacteria. Centrifugation and washing were performed either once or twice, depending on the sample. After centrifugation, the supernatant was removed from the samples at each wash stage (unwashed, single wash, and double wash), and the pellet was resuspended by agitation with a vortex mixer. Then, 3 mL of 2% (v/v) glutaraldehyde solution was added, and the mixture was stirred gently. The samples were incubated at room temperature for a minimum of 24 hours. The sample was then centrifuged at 8,000 × g for 2 minutes using a swinging rotor. The resulting pellets were resuspended in 1 mL of 0.1 M phosphate buffer (PBS, pH 7.4) and centrifuged again under the same conditions. The pellets were resuspended in 1 mL of distilled water, allowed to stand at room temperature for 20 minutes, and centrifuged again. This washing process was repeated twice. Dehydration was performed using an ethanol concentration gradient. The pellets were sequentially resuspended in 1 mL of 50%, 70%, 80%, and 90% (v/v) ethanol and soaked for 10 minutes at room temperature. Subsequently, two 10-minute treatments with 99.5% (v/v) ethanol were performed, each followed by centrifugation and removal of the supernatant. The ethanol was then replaced with t-butanol. First, the samples were resuspended in 1 mL of a 1:1 mixture of ethanol and t-butanol, incubated for 10 minutes at room temperature, and centrifuged at 8,000 × g for 5 minutes. This process was repeated twice. The pellets were resuspended twice in pure t-butanol, and the supernatant was removed each time. Finally, the bacterial samples were soaked in t-butanol, frozen, and subsequently lyophilized. The dried samples were seeded with gold prior to SEM observation using a JCM-6000Plus NeoScope (JEOL Ltd., Tokyo, Japan).

### Flow cytometric analysis

The absorbance of *P. copri* JCM 13464[T] cultures incubated overnight at 37°C in BG medium was measured at 660 nm. One milliliter of the culture was transferred into 1.5 mL tubes and divided into three groups: unwashed, single-washed, and double-washed with sterile distilled water. Samples from each group were centrifuged at 8,000 × g for 10 minutes at 4°C, and the supernatant was removed. For samples other than those in the unwashed group, 1 mL of sterile distilled water was added, and the pellets were resuspended using a vortex mixer. For the double-washed group, the washing step was

performed twice in total. Finally, the pellets from all groups were resuspended in 1 mL of PBS and kept on ice. Each bacterial suspension was adjusted to an optical density of 0.1 at 660 nm, followed by centrifugation of 1 mL at 8,000 × g for 10 minutes at 4°C to remove the supernatant. To the pellet, 100 μL of PBS or 100 μL of propidium iodide solution (0.1 mg/mL [Merck KGaA, Darmstadt, Germany] diluted in PBS) was added and the mixture was left on ice for 10 minutes. Then, 1 mL of PBS was added to all the samples and centrifuged at 8,000 × g for 10 minutes at 4°C. After this washing process was repeated three times, the samples were finally resuspended in 250 μL of PBS and subjected to flow cytometric analysis. Flow cytometric analysis was conducted using a MACSQuant Analyzer 16 (Miltenyi Biotec, Bergisch Gladbach, Germany) and analyzed with FlowLogic V7.1 software (Inivai Technologies, Mentone, Victoria, Australia).

## Growth and survival assay of bacteria

For the proliferation test, *P. copri* JCM 13464$^T$ culture was inoculated at 1% (v/v) in BG or TJ medium and incubated at 37°C for 15 hours under anaerobic conditions. Prior to the proliferation assay, the growth curve of *P. copri* JCM 13464$^T$ was evaluated, confirming that 15 hours of incubation corresponded to the late logarithmic phase. Absorbance was measured at 660 nm, 15 hours after the start of the culture.

In the post-NGM viability test, NGM plates seeded with *P. copri* JCM 13464$^T$, *M. funiformis* JCM 14723$^T$, *S. salivarius* JCM 5707$^T$, or *E. coli* OP50-were opened on a clean bench, 2 mL of BG medium was added, and the pellets were gently tapped with a Conradi rod to suspend the bacteria. The suspension was collected in a 15 mL tube, and the volume of the BG medium was adjusted to 10 mL. The sample was agitated with a vortex mixer and the absorbance at 660 nm was measured. The bacterial suspension was then inoculated into 10 mL of fresh BG medium to achieve a final absorbance of 0.02 at 660 nm. Four tubes were prepared per group and were anaerobically cultured at 37°C. Samples were opened on days 1 (before the start of incubation), 2, 3, and 4, and absorbance was measured at 660 nm.

## LC-MS/MS analysis

As described in the 'Preparation of NGM plates with test bacteria,' *P. copri* JCM 13464$^T$ was cultured in BG or TJ medium, filtered through a cell strainer, and then centrifuged at 8,000 × g for 10 minutes at 4°C. The supernatant was collected in its entirety (supernatant sample). As negative controls, uninoculated BG and TJ media were filtered and centrifuged in the same manner, and the supernatant was collected (medium sample). The collected supernatant and medium samples were stored at −80°C. The remaining pellet was suspended in 40 mL of sterile distilled water and centrifuged at 8,000 × g for 10 minutes at 4°C to remove the supernatant. This washing procedure was repeated three times. All supernatants were removed, and the pellet was stored at −80°C. The bacterial samples for metabolomic analysis were prepared according to a previously reported method [17]. Specifically, 5 mL of chilled methanol pre-cooled to −80°C for at least 30 minutes was rapidly added to the bacterial pellet taken from the −80°C freezer, and the mixture was suspended using a vortex mixer. Liquid nitrogen was poured into a 500 mL beaker set on ice, and the methanol suspension of the bacteria was refrozen by immersing it in liquid nitrogen. The frozen bacterial suspension was transferred to ice and thawed. This freeze-thaw cycle was performed three times. The supernatant was collected in 1.5 mL microtubes after centrifugation at 8,000 × g for 10 minutes at 4°C. The collected supernatant was further centrifuged at 14,000 × g for 10 minutes at 4°C and used as the bacterial sample for metabolomic analysis. All samples were filtered (hydrophilic PTFE membrane, 0.45 μm, Shimadzu Corporation, Kyoto, Japan) and then transferred to HPLC vials. Standard curves for quantifying the concentration of L-(-)-3-PLA in culture supernatants were prepared by dissolving L-(-)-3-PLA (Tokyo Chemical Industry Co., Ltd., Tokyo, Japan) in methanol at concentrations of 1, 5, 10, 50, and 100 μM. Similarly, standard curves for bacterial samples were prepared by dissolving L-(-)-3-PLA in methanol at concentrations of 0.00064, 0.0032, 0.016, 0.08, 0.4, 2, 10, and 50 μM. These samples were also filtered and transferred to HPLC vials.

Samples were analyzed using an ultra-high-performance liquid chromatography (UHPLC) system (UltiMate 3000; Thermo Fisher Scientific Inc., Waltham, MA, USA). A TSKgel ODS-100V column (5 μm, 3 × 50 mm; Tosoh Corporation,

Tokyo, Japan) was used for separation and was coupled to a Q Exactive Orbitrap mass spectrometer (Thermo Fisher Scientific, Inc.). The mass spectrometer was operated at a resolution of 17,500 in both positive and negative ion modes, with a detection range of 150–1,500 m/z. The LC mobile phase consisted of (A) distilled water (KANTO CHEMICAL Co., Inc., Tokyo, Japan) containing 0.1% (v/v) formic acid (Honeywell International Inc., Tokyo, Japan) and (B) acetonitrile (KANTO CHEMICAL Co., Inc.) containing 0.1% (v/v) formic acid. A multistep gradient system was employed: 3% B for the first 10 minutes, followed by a linear gradient from 3% to 97% B over 20 minutes, and maintained at 97% B for an additional 5 minutes. The flow rate was 0.4 mL/minute. The analysis time was 35 minutes, and the injection sample volume was 5 μL. Compound Discoverer 3.2 (Thermo Fisher Scientific, Inc.) was used to annotate the components obtained from the MS/MS analysis. As in a previous report [18], only components with mzCloud best-match confidence score greater than 70% were used among the detected components. When multiple components with the same name were detected, data with the highest mzCloud best-match confidence score were used. If multiple peaks were detected for the same component with the same name and mzCloud best-match confidence score, they were assumed to represent a single component; therefore, the combined value was treated as the area of that component. The area ratio of TJ medium-cultured bacteria to BG medium-cultured bacteria was calculated, and based on previous reports [19,20], a $\log_2$ fold change of ±1 was set as the threshold for a significant difference. In addition, technical errors in metabolomic analyses have been reported [21], with a median error estimated to be up to 20%. Applying a 2-fold safety factor and accounting for a technical error of 40%, a $\log_2$ fold change of ±0.49 was considered to indicate a minor variation in peak area ratios.

The concentration of L-(-)-3-PLA in each sample was calculated based on the standard curve. For bacterial samples, the concentration was converted to the amount contained in 50 μL of the bacterial suspension applied to NGM. Specifically, the L-(-)-3-PLA concentration (μM) in the methanol extract, as determined using the standard curve, was used to calculate the amount of L-(-)-3-PLA in the total volume (5 mL) of the bacterial extract. Since 50 μL of bacterial suspension can be obtained 20 times from 100 mL of culture medium, the amount of L-(-)-3-PLA in the 5 mL extract was divided by 20 to calculate the amount of L-(-)-3-PLA in 50 μL of bacterial suspension.

## Statistical analysis

In the Smurf assay, Fisher's exact probability test was used for group comparisons, with *P*-values adjusted by the Holm method, as previously reported [22,23]. For all other assays, tests for homogeneity of variances were conducted prior to the group comparisons. For two-group comparisons, variance equality was assessed using the *F*-test; for three-group comparisons, the Bartlett test was applied. When a potential violation of the equal variance assumption was detected ($P < 0.1$), Welch's *t*-test was used for both two- or three-group comparisons. In the case of three-group comparisons, *P*-values were adjusted using the Holm method. When the equal variance assumption was considered not violated ($P \geq 0.1$), Student's *t*-test was used for two-group comparisons, and Dunnett's test for three-group comparisons. All statistical tests were performed as two-tailed tests using EZR version 1.40 (Saitama Medical Center, Jichi Medical University, Saitama, Japan) [24]. A *P*-value less than 0.05 was considered statistically significant.

## Results

### Cultivation of *P. copri* JCM 13464[T] in TJ medium suppresses its intestinal permeability-enhancing effect on *C. elegans*

*E. coli* OP50, *P. copri* JCM 13464[T], *M. funiformis* JCM 14723[T], and *S. salivarius* JCM 5707[T] were grown anaerobically in BG or TJ medium, concentrated, and added to NGM. *C. elegans* were fed *E. coli* OP50, the standard diet, for 4 days after hatching, and then fed the test organisms applied to NGM from days 5–8. On day 8, intestinal permeability of *C. elegans* was evaluated using the Smurf assay (Fig 1A). For *P. copri* JCM 13464[T], the proportion of Smurfs was significantly lower in *C. elegans* that had ingested bacteria grown on the TJ medium than in those grown on the BG medium (Fig 1B). For *E. coli*, *M. funiformis* and *S. salivarius*, no significant differences were observed between the two media. No significant

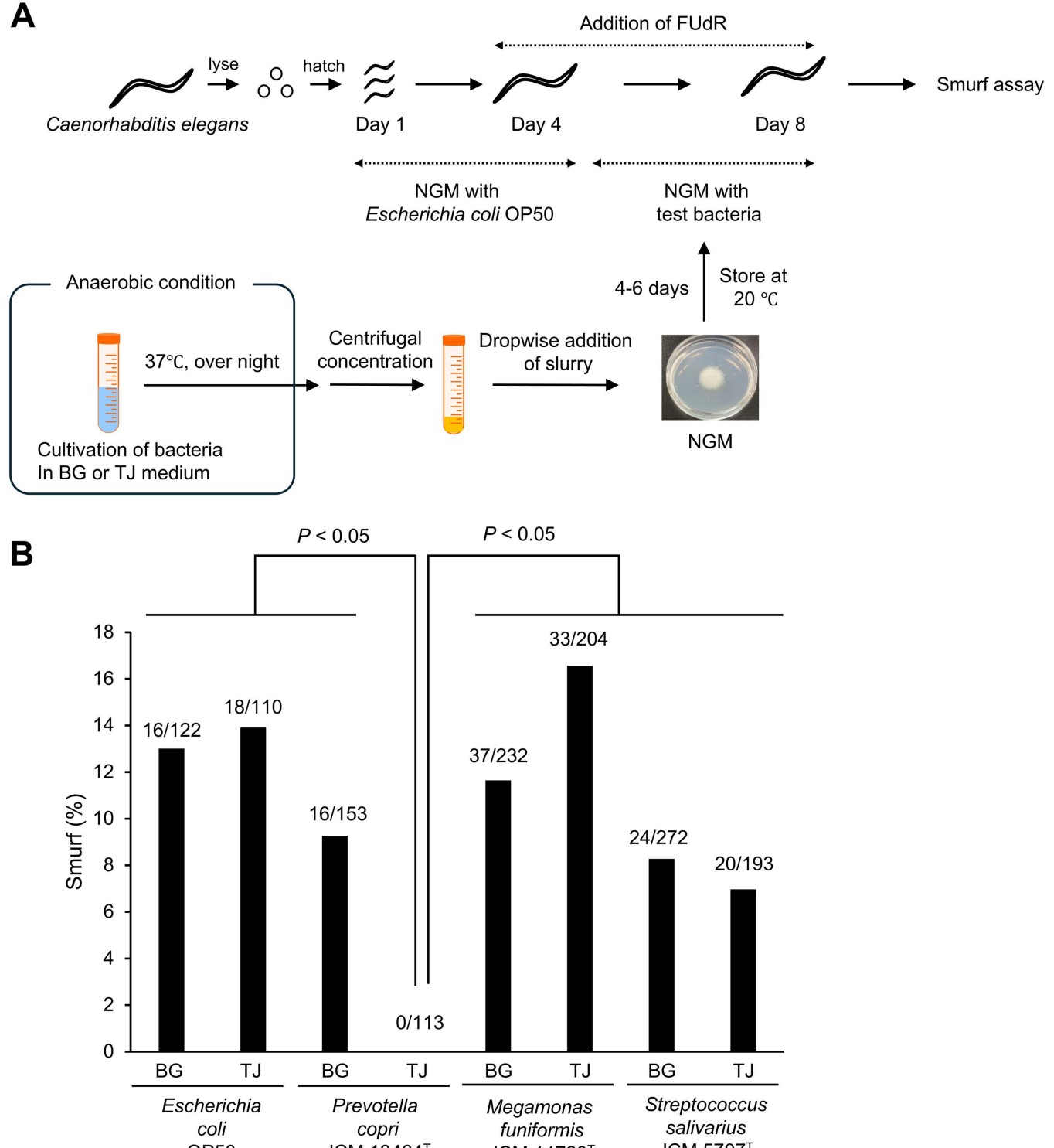

**Fig 1. Effect of intestinal bacteria cultured in BG medium or BG medium supplemented with tomato juice on the intestinal permeability of** *Caenorhabditis* **elegans.** (A) Hatched *Caenorhabditis* (*C.*) *elegans* were reared on *Escherichia coli* OP50-seeded nematode growth medium (NGM) for 4 days, and then transferred to NGM medium seeded with different intestinal bacteria for an additional 3 days (reared for a total of 8 days). Intestinal

bacteria were grown overnight anaerobically in BG medium or BG medium with 1% (v/v) tomato juice (TJ), centrifuged, concentrated, and then applied onto NGM. After storage at 20°C for 4−6 days, the bacteria were used to feed *C. elegans*. During the *C. elegans* feeding period, NGM with intestinal bacteria was replaced daily. On the 8th day of *C. elegans* rearing, the Smurf assay was performed and Smurf individuals were counted under a microscope. FUdR: 5-Fluoro-2'-deoxyuridine. (B) Results of the Smurf assay. The numbers at the top of the bar graph represent the actual number of Smurf individuals among all individuals analyzed. The results of the two tests were combined and the percentages of Smurf individuals were compared between groups using the Holm-adjusted Fisher's exact probability test.

difference in the proportion of Smurf individuals was observed between *E. coli* OP50 and *P. copri*, *M. funiformis,* or *S. salivarius* cultured on BG medium.

### The increase in intestinal permeability caused by *P. copri* JCM 13464ᵀ is due to live bacteria

To investigate the mechanism by which the culture in TJ medium suppressed the increase in intestinal permeability caused by *P. copri* JCM 13464ᵀ, we first examined whether the bacteria or culture supernatant were responsible for the increase or suppression of intestinal permeability. Specifically, bacteria cultured in BG or TJ media were washed twice with sterile distilled water, resuspended in the original culture supernatant or other culture supernatants, and fed to *C. elegans*. The proportion of Smurf individuals was significantly lower in the group fed bacteria cultured in BG medium, washed with distilled water, and resuspended in the original culture supernatant than in the group fed unwashed BG medium-cultured bacteria (Fig 2, BG cells+BG sup vs. BG [unwashed]). Smurf proportions did not differ significantly between the following comparisons (Fig 2): BG cells+BG sup vs. BG cells+TJ sup, unwashed TJ medium-cultured bacteria vs. TJ cells+TJ sup, and TJ cells+TJ sup vs. TJ cells+BG sup. Based on these results, it was hypothesized that the bacteria themselves, and not the culture supernatant,

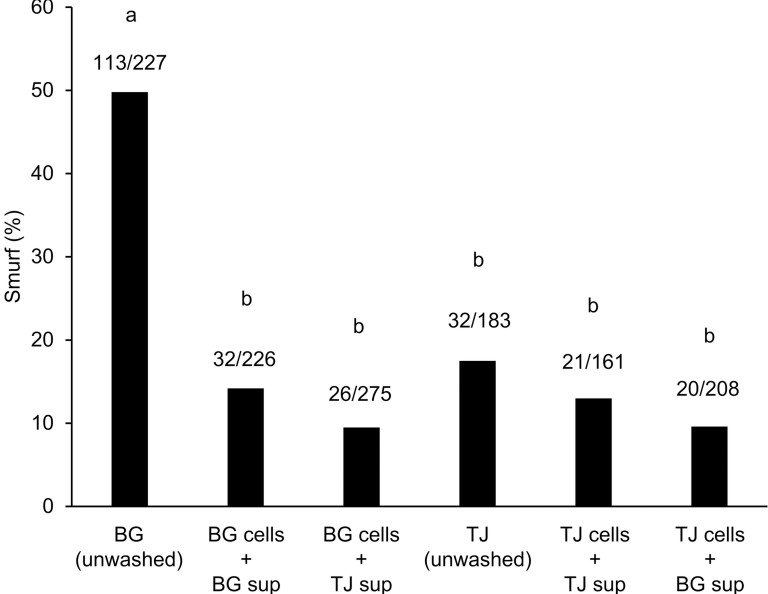

**Fig 2. Localization of components involved in intestinal permeability of *Prevotella copri* JCM 13464ᵀ.** *Prevotella copri* JCM 13464ᵀ was cultured overnight in BG medium or BG medium supplemented with 1% (v/v) tomato juice (TJ medium), and a portion of the culture was centrifuged, concentrated, and addedto nematode growth medium (NGM) (unwashed samples). The remaining bacterial cells were washed with sterile distilled water, suspended in the original culture supernatant or an alternative culture supernatant, and then applied to the NGM. The Smurf assay was performed, as shown in Fig 1. In the graph, "cell" indicates the bacterial cells, and "sup" indicates the origin of the culture supernatant. The numbers at the top of the bar graphs represent the actual measured values for Smurf individuals among all individuals analyzed. The results of the two tests were combined and the proportions of Smurf individuals were compared between groups using the Holm-adjusted Fisher's exact probability test. A significant difference was observed between the different signs ($P < 0.001$).

Wash with distilled water

| Unwashed | single | double |
|---|---|---|

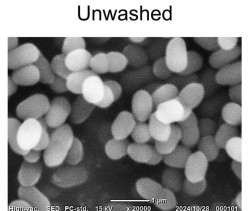 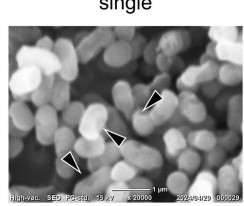 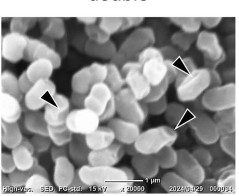

**Fig 3. Scanning electron microscopy (SEM) images of *Prevotella copri* JCM 13464^T washed with sterile distilled water.** *Prevotella copri* JCM 13464^T was cultured overnight in BG medium, and part of the culture was washed once or twice with sterile distilled water. After centrifugation and removal of the supernatant, the bacterial cells were fixed and the morphology of the bacteria was observed using a scanning electron microscope. In the image, the arrowheads indicate indentations in the bacterial cells, and the white lines indicate 1 μm. This is a representative image from two experiments.

were responsible for the intestinal permeability of *C. elegans*. It was also speculated that these bacteria would lose their ability to increase the intestinal permeability of *C. elegans* when washed with sterile distilled water.

To confirm the changes in bacterial cells when *P. copri* JCM 13464^T was washed with sterile distilled water, the cells were examined using a scanning electron microscope before and after being washed once or twice with sterile distilled water. Large indentations were observed on the surface of the cells that had been washed with sterile distilled water compared to the unwashed cells (Fig 3).

It has been reported that in *E. coli*, exposure to a hypotonic environment causes an influx of water into the cytoplasm, which in turn activates mechanosensitive channels [25]. This leads to the release of water and ions into the periplasmic space, resulting in its expansion and causing the outer membrane to display surface undulations and localized indentations, which are similar to those seen in *P. copri* JCM 13464^T in Fig 3. To investigate whether osmotic stress might affect *P. copri* JCM 13464^T viability under our experimental conditions, the bacteria were stained with propidium iodide (PI) and subjected to flow cytometry analysis after being left unwashed or washed once or twice with sterile distilled water. After extracting the data for the bacterial cells based on forward scatter (FSC) and side scatter (SSC), the bacteria were classified into four subpopulations based on the intensities of the FSC and PI. Washing the bacterial cells with sterile distilled water significantly reduced the proportion of bacteria in the FSC$^{high}$PI$^-$, FSC$^{high}$PI$^+$, and FSC$^{low}$PI$^+$ fractions, and significantly increased the proportion of bacteria in the FSC$^{low}$PI$^-$ fraction (Fig 4, S3 Table).

As previously reported [26,27], the FSC$^{low}$PI$^-$ fraction is thought to be bacterial debris. To investigate the involvement of live *P. copri* JCM 13464^T in intestinal permeability, we fed live or heat-killed *P. copri* JCM 13464^T to *C. elegans* and performed the Smurf assay. The proportion of Smurf individuals among the heat-killed bacteria was significantly lower than that among the live bacteria in the BG medium-cultured group (Fig 5). For bacteria cultured in TJ medium, no difference was observed in the proportion of Smurf individuals between live and heat-killed samples.

### Cultivation in TJ medium significantly reduces the survival rate of *P. copri* JCM 13464^T

The turbidity of the culture fluid of *P. copri* JCM 13464^T was measured after 15 hours of incubation in BG or TJ medium (i.e., before application to the NGM). There was no significant difference in the turbidity of the culture fluid between BG and TJ media (Fig 6A). In contrast, the bacteria applied to NGM and stored at 20°C for 4 days (i.e., the same conditions as those used for ingestion by *C. elegans* as in Fig 1A) were resuspended in BG medium. After adjusting for turbidity, the bacterial suspension was incubated at 37°C under anaerobic conditions for three days. The turbidity of the bacteria grown on TJ medium was significantly lower than, or tended to be lower than that of the bacteria grown on BG medium from the second day of incubation (Fig 6B, S4 Table).

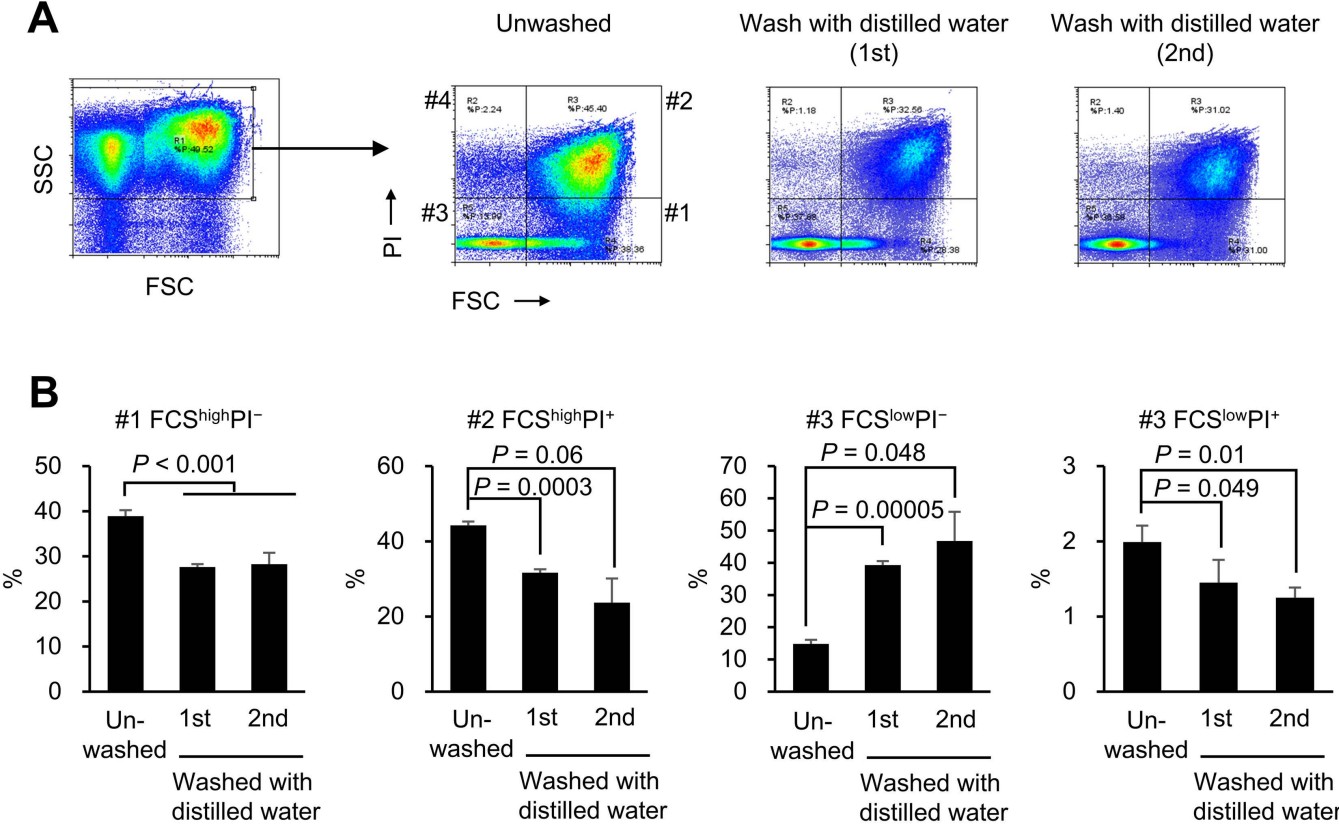

**Fig 4. Flow cytometric analysis of *Prevotella copri* JCM 13464ᵀ washed with sterile distilled water.** *Prevotella copri* JCM 13464ᵀ was cultured overnight in BG medium and a portion was washed once or twice with sterile distilled water. After centrifugation and removal of the supernatant, the bacterial cells were stained with propidium iodide (PI) and analyzed using a flow cytometer. (A) The gating procedure for flow cytometry analysis. First, we extracted bacterial data based on the forward scatter (FSC) and side scatter (SSC). The post-extraction data were further processed based on the fluorescence intensity of FSC and PI, and classified into four subpopulations based on high and low FSC values and positive and negative PI staining. A representative dot plot is shown for the three experiments. (B) The percentages of each subpopulation of unwashed bacteria or bacteria washed once or twice with sterile distilled water are summarized. The bar graph shows the mean, and the error bars indicate the standard deviation. The results of the Bartlett test of homogeneity of variances indicated significant differences for both FCS^highPI^+ ($P=0.03$) and FCS^lowPI^- ($P=0.02$). Therefore, a Welch's *t*-test with *P*-value adjustment using the Holm method was used to compare the three groups. For all other data, Dunnett's test was performed (vs. unwashed). ($n=3$).

As shown in Fig 1A, the effect of bacterial cultivation in TJ medium on intestinal permeability varied among the bacterial strains, with only *P. copri* JCM 13464ᵀ showing a decrease in the proportion of Smurf individuals among the four tested strains. We assessed whether this phenomenon was related to the bacterial survival rate following NGM application. The method was the same as that in Fig 6B; however, for the culture duration after recovery of the bacteria from NGM, the turbidity change was measured until the following day, as in the actual *C. elegans* cultivation, where new plates are used after one day. As a result, *M. funiformis* JCM 14723ᵀ showed a decrease in turbidity in both BG and TJ medium groups, but no significant difference between groups was observed (Fig 6C, S4 Table). Both *S. salivarius* JCM 5707ᵀ and *E. coli* OP50 exhibited an increase in turbidity in both BG and TJ medium groups. For *S. salivarius* JCM 5707ᵀ, the increase in turbidity was suppressed in the TJ medium group compared to the BG medium group, while no significant difference was observed between BG and TJ medium groups for *E. coli* OP50.

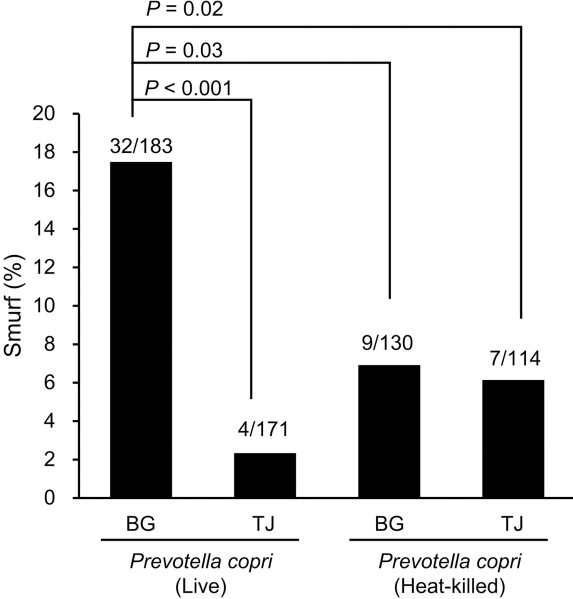

**Fig 5. Effect of heat inactivation on changes in intestinal permeability caused by *Prevotella copri* JCM 13464[T].** *Prevotella copri* JCM 13464[T] was cultured overnight in BG medium or BG medium supplemented with 1% (v/v) tomato juice (TJ medium), concentrated by centrifugation, and applied to nematode growth medium (NGM) ("Live" sample). After centrifugation and concentration, a portion of the culture fluid was heated at 60°C for 30 minutes and then applied to NGM ("heat-killed" sample). The Smurf assay was then performed as shown in Fig 1. The numbers at the top of the bar graph represent the actual measured values for Smurf individuals in all individuals analyzed. The proportions of Smurf individuals were compared between groups using the Holm-adjusted Fisher's exact probability test. *P*-values are shown only for comparisons between groups with significant differences.

## Preliminary metabolomic analysis of *P. copri* JCM 13464[T] cultured in BG medium and TJ medium

The BG medium, TJ medium, and culture supernatants of *P. copri* JCM 13464[T] cultured in these media, as well as the extracts of the cultured bacteria, were analyzed using LC-MS/MS. Of the detected substances, 44 had an mzCloud best match confidence score of 70 or higher. The quantitative changes in these substances due to incubation in the TJ medium were assessed in bacteria suspected to be involved in modulating intestinal permeability. A $\log_2$ fold-change exceeding ±1 was interpreted as a quantitative change. The results showed that L-(-)-3-PLA and *trans*-3-indoleacrylic acid were detected at higher levels in TJ medium-cultured bacteria than in BG medium-cultured bacteria, with $\log_2$ fold change values of +2.05 and +1.81, respectively (Fig 7). In contrast, Ile-Leu-Lys, 2,3,4,9-tetrahydro-1H-β-carboline-3-carboxylic acid, Cyclo(phenylalanyl-prolyl), 3-(propan-2-yl)-octahydropyrrolo[1,2-a]pyrazine-1,4-dione, and NP-011220 were present at lower levels in TJ medium-cultured bacteria than in BG medium-cultured bacteria, with $\log_2$ fold-change values of −1.46, −1.50, −2.82, −4.66 and −4.67, respectively.

For L-(-)-3-PLA, a comparison between BG and TJ media showed no clear difference in its concentration in either medium ($\log_2$ fold-change = −0.33, S5 Table). In contrast, a slightly higher amount of L-(-)-3-PLA was detected in the supernatant of the TJ medium culture than in that of the BG medium culture ($\log_2$ fold change = 0.9; S5 Table). L-Phenylalanine, the precursor of L-(-)-3-PLA, was detected at slightly higher levels in the TJ medium than in the BG medium in the uninoculated medium ($\log_2$ fold change = +0.8, S5 Table), however, this difference was not observed in the culture supernatant or in the bacteria after incubation ($\log_2$ fold change = −0.10 and −0.37, respectively; S5 Table).

## The effects of L-(-)-3-PLA on the vulnerability of *P. copri* JCM 13464[T] and intestinal permeability in *C. elegans*

To evaluate the impact of L-(-)-3-PLA on the vulnerability of *P. copri* JCM 13464[T] and its effect on intestinal permeability in *C. elegans*, the concentration of L-(-)-3-PLA in the TJ culture supernatant of *P. copri* JCM 13464[T]

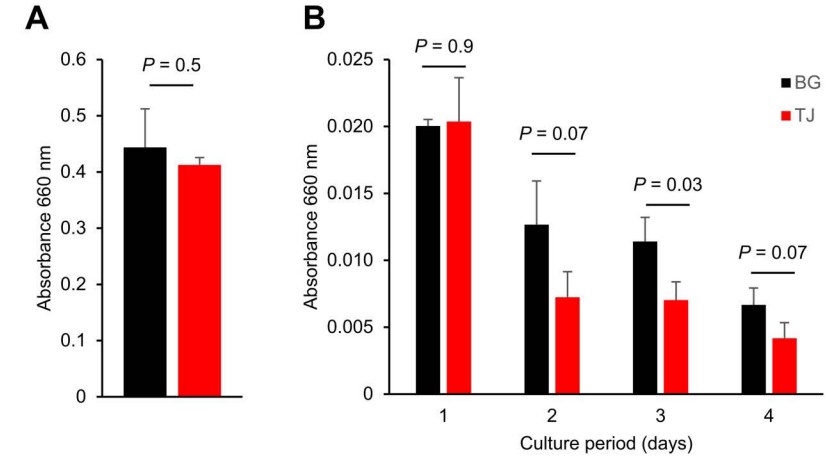

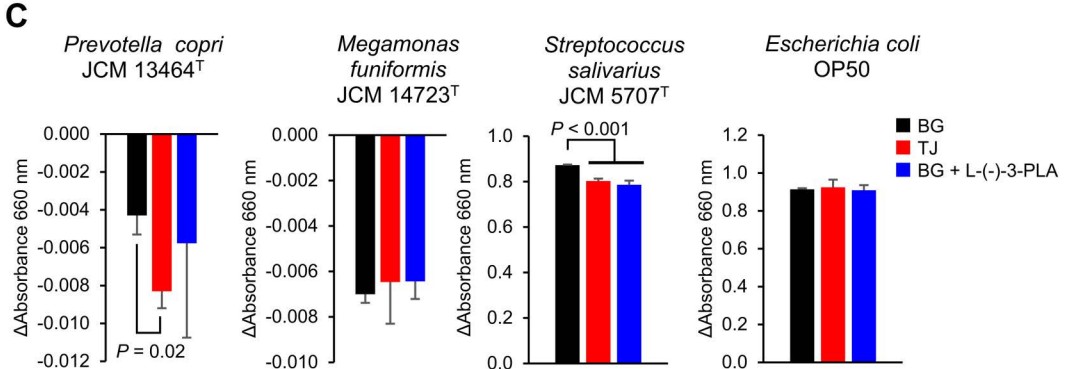

**Fig 6. Effect of culturing in BG medium supplemented with tomato juice or L-(-)-3-phenyllactic acid on the growth or survival rate of the tested bacterial strains.** (A) *Prevotella copri* JCM 13464[T] was cultured for 15 hours in BG medium or BG medium supplemented with 1% (v/v) tomato juice (TJ medium), and the absorbance at 660 nm was compared between groups. (B) *Prevotella copri* JCM 13464[T] was cultured overnight in BG or TJ medium, concentrated by centrifugation, and applied to nematode growth medium (NGM). After storing it at 20°C for 4 days, the bacteria applied to the NGM were resuspended in BG medium, and after adjusting the turbidity, they were anaerobically cultured at 37°C for 4 days. Absorbance at 660 nm was measured from the first day of incubation (before the start of incubation) to the fourth day. (C) The experiment was conducted as described for (B), with the following modifications. In addition to *Prevotella copri* JCM 13464[T], *Escherichia coli* OP50, *Megamonas funiformis* JCM 14723[T], and *Streptococcus salivarius* JCM 5707[T] were also tested. The bacteria were cultured overnight in BG or TJ medium, with a group receiving BG medium supplemented with 5.5 μM of L-(-)-3-phenyllactic acid (L-(-)-3-PLA). After centrifugation and application to NGM, the bacteria were stored at 20°C for 4 days. Following resuspension in BG medium and adjustment of turbidity, the cultures were anaerobically incubated at 37°C for 2 days. Absorbance at 660 nm was measured from the first day (before the start of incubation) to the second day. Data are presented as the change in absorbance at 660 nm from day 1 to day 2. (A-C) The bar graph shows the means and the error bar shows the standard deviations. The results of an *F*-test or the Bartlett test of homogeneity of variances indicated significant differences between the BG and TJ groups on day 1 of culture in Fig 6B (*P* = 0.04), and also between the groups in *Prevotella copri* JCM 13464[T] in Fig 6C (*P* = 0.05). Therefore, Welch's *t*-tests were used for comparisons between two groups, and Welch's *t*-tests with *P*-value adjustment using the Holm method were used for comparisons between three groups. For all other data, Student's *t*-tests or Dunnett's tests (vs. BG group) were performed for two- or three-group comparisons, respectively. (n = 3).

and in the bacterial suspension applied to NGM was calculated. The results showed that the concentration of L-(-)-3-PLA in the TJ culture supernatant of *P. copri* JCM 13464[T] was 5.5 μM, whereas the concentration of L-(-)-3-PLA in the TJ medium-cultured *P. copri* JCM 13464[T] bacterial suspension applied to NGM was 10 nM (0.5 pmol per 50 μL).

To evaluate the impact of L-(-)-3-PLA on the vulnerability of the bacterial strains, each strain was cultured in BG medium with or without supplementation of 5.5 μM L-(-)-3-PLA. The bacterial survival rate following application to NGM was then assessed. *S. salivarius* JCM 5707[T] exhibited a significantly smaller change in turbidity in the BG medium

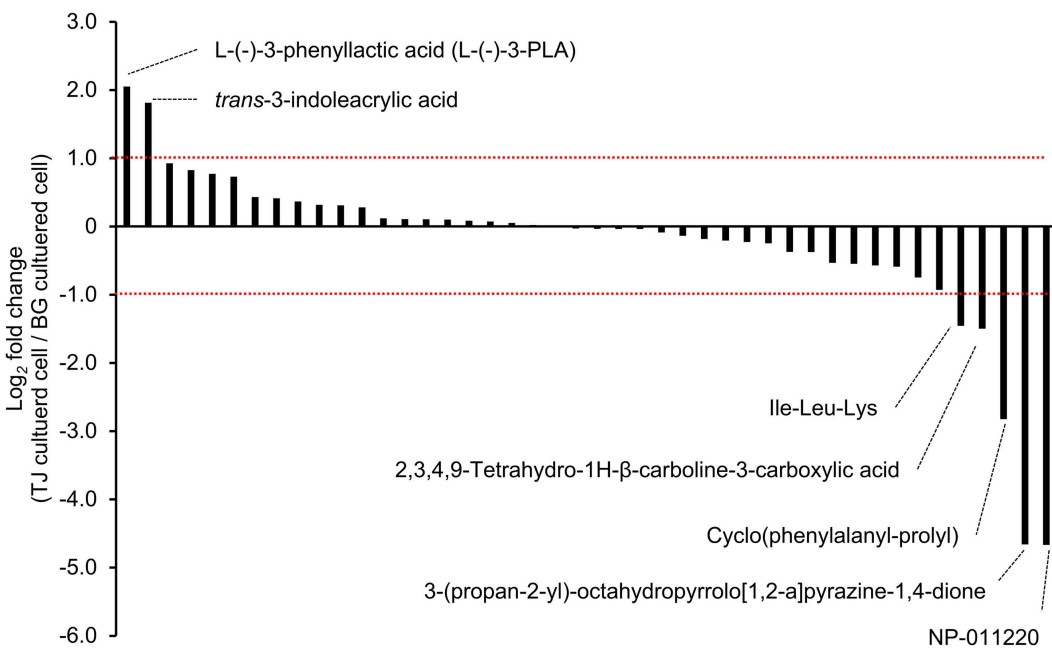

**Fig 7. Metabolomic analysis of *Prevotella copri* JCM 13464ᵀ bacteria cultured in BG medium and BG medium containing tomato juice.**
*Prevotella copri* JCM 13464ᵀ was grown overnight in BG medium or BG medium with 1% (v/v) tomato juice (TJ medium). The bacteria were collected and methanol-extracted samples were subjected to liquid chromatography-tandem mass spectrometry (LC-MS/MS) analysis. Area ratios ($\log_2$ fold change) between BG and TJ media-cultured bacteria were calculated for substances detected with a mzCloud best-match confidence of at least 70%. The graphs are arranged from left to right in order of increasing area-to-area ratio. The red dashed line indicates the threshold where the $\log_2$ fold change of the area ratio is ± 1. For substances that exceeded the threshold, the name of the component that was the top hit in the analysis software Compound Discoverer 3.2 is described. (n = 1).

supplemented with L-(-)-3-PLA compared to the control group cultured in BG medium alone. No significant differences in turbidity changes between groups were observed for *P. copri* JCM 13464ᵀ, *M. funiformis* JCM 14723ᵀ, and *E. coli* (Fig 6C).

The potential direct impact of L-(-)-3-PLA on intestinal permeability in *C. elegans* was evaluated. *C. elegans* were cultured on NGM plates containing 0, 10, or 100 nM of L-(-)-3-PLA (with *E. coli* OP50 as the food source), and the Smurf assay was conducted. The results showed no significant differences in the proportion of Smurf individuals between any of the three groups (Fig 8). However, a significant difference in the proportion of Smurf individuals was observed when comparing the control group (0 nM) with the 10 nM group, which corresponds to the concentration found in bacterial samples cultured in TJ medium ($P = 0.04$, Fig 8, dashed line).

## Discussion

In this study, we evaluated the effects of tomato consumption on *Prevotella*, *Megamonas*, and *Streptococcus*, as well as the increased intestinal permeability induced by these bacteria, using *C. elegans*. As noted in the Introduction, *Prevotella*, *Megamonas*, and *Streptococcus* are gut bacterial genera implicated in increased intestinal permeability in human epidemiological studies. Therefore, we compared the intestinal permeability of *C. elegans* fed these bacteria with that of worms fed *E. coli* OP50, the standard laboratory diet for *C. elegans*. However, no significant differences were observed in the proportion of "Smurf" phenotype individuals between worms fed *E. coli* OP50 and those fed *P. copri* JCM 13464ᵀ, *M. funiformis* JCM 14723ᵀ, or *S. salivarius* JCM 5707ᵀ. *E. coli* OP50 is commonly used as the standard food source for *C. elegans* because it grows slowly and forms a thin bacterial lawn on NGM agar plates, facilitating microscopic observation of the worms. Moreover, it is well-suited for laboratory use due to its safety and ease of cultivation. Nonetheless,

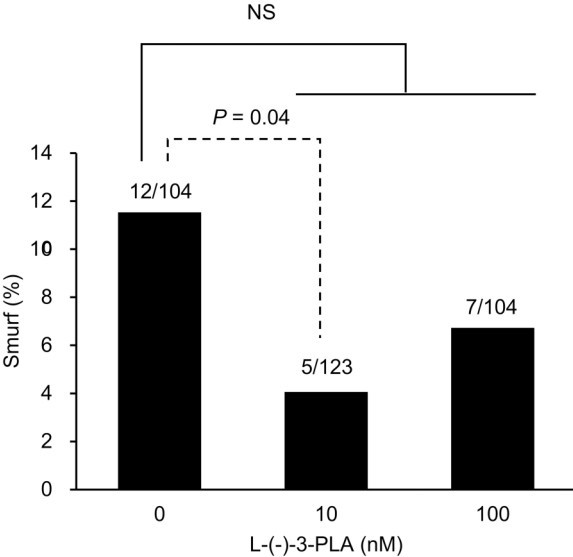

**Fig 8. Effect of L-(-)-3-phenyllactic acid on intestinal permeability in *Caenorhabditis elegans*.** *Escherichia coli* OP50 was cultured overnight in BG medium, concentrated by centrifugation, and applied to nematode growth medium (NGM) supplemented with 0, 10, or 100 nM L-(-)-3-phenyllactic acid (L-(-)-3-PLA). The Smurf assay was conducted as described in Fig 1. The numbers above each bar in the graph indicate the actual count of Smurf individuals among all worms analyzed. The proportions of Smurf individuals were compared between groups using the Holm-adjusted Fisher's exact probability test. Additionally, a separate comparison between the 0 nM and 10 nM L-(-)-3-PLA groups was performed using the unadjusted Fisher's exact probability test. *P*-values are shown only for comparisons between groups with significant differences. NS, not significant.

wild *C. elegans* harbor diverse gut microbiota. Previous studies have shown that feeding *C. elegans* a bacterial mixture resembling the natural gut microbiota of wild nematodes can extend lifespan compared to worms fed *E. coli* OP50 [28]. This suggests that *E. coli* OP50 may not be the optimal diet for maintaining gut barrier function. Therefore, this study does not focus on direct comparisons between *E. coli* OP50 and the other tested bacterial strains. Instead, we focused on the effects of culturing each bacterial species in TJ medium, emphasizing their individual impact on host intestinal permeability.

Among the three bacterial strains, *P. copri* JCM 13464[T], *M. funiformis* JCM 14723[T], and *S. salivarius* JCM 5707[T], only *P. copri* JCM 13464[T] exhibited an inhibitory effect on intestinal permeability when cultured in TJ medium. As part of our investigating into the underlying mechanism, we compared the survival rates of each bacterial strain following NGM application. The results showed that a decrease in turbidity, indicative of bacterial lysis, was observed only for *P. copri* JCM 13464[T] and *M. funiformis* JCM 14723[T]. However, unlike *P. copri* JCM 13464[T], *M. funiformis* JCM 14723[T] did not exhibit any significant difference in the extent of lysis between the BG and TJ medium groups. For *S. salivarius* JCM 5707[T], a decrease in turbidity was observed in the TJ medium group compared to the BG medium group; however, turbidity increased in both groups from day 1, indicating bacterial growth. Therefore, the lower turbidity in the TJ medium group for *S. salivarius* JCM 5707[T] was considered to reflect growth inhibition rather than enhanced lysis. Furthermore, in the NGM application of this study, the plates were replaced daily. Therefore, it is assumed that the growth inhibition of *S. salivarius* JCM 5707[T] by TJ medium does not affect the interpretation of Fig 1. These results suggest that the reduction in intestinal permeability induction by cultivation in TJ medium observed only in *P. copri* JCM 13464[T] is associated with the ease of bacterial lysis following NGM application.

L-(-)-3-PLA and *trans*-3-indoleacrylic acid, which exhibited increased intracellular accumulation in TJ medium cultivation, as well as Ile-Leu-Lys, 2,3,4,9-Tetrahydro-1H-β-carboline-3-carboxylic acid, Cyclo(phenylalanyl-prolyl), 3-(propan-2-yl)-octahydropyrrolo[1,2-a]pyrazine-1,4-dione, and NP-011220, which showed decreased intracellular

concentrations under TJ medium cultivation conditions, were hypothesized to be bioactive metabolites. Although previous studies have reported that *trans*-3-indoleacrylic acid inhibits cyanobacterial growth [29], there is no direct evidence of its potential antimicrobial activity. The components with decreased content in the bacteria were discussed in terms of their necessity for *P. copri* JCM 13464[T] survival: Cyclo(phenylalanyl-prolyl) is an antimicrobial substance [30], and the relationship between its decrease and *Prevotella* viability is not known, nor have previous studies investigated the effects of Ile-Leu-Lys, 2,3,4,9-Tetrahydro-1H-β-carboline-3-carboxylic acid, 3-(propan-2-yl)-octahydropyrrolo[1,2-a]pyrazine-1,4-dione, or NP-011220 on *Prevotella* viability.

In contrast, PLA exhibited a wide range of antibacterial activities. Specifically, it has been reported to increase the permeability of the outer membrane and inhibit DNA replication in *E. coli*, a Gram-negative bacterium, similar to *Prevotella* [31]. Furthermore, it has been suggested that the antibacterial activity of PLA is enhanced when combined with oxidative stress [32,33]. In the present study, there was no difference in turbidity between cultures on BG and TJ media during anaerobic cultivation of *P. copri* JCM 13464[T] in liquid medium, and a decrease in turbidity in TJ-cultured bacteria after application to NGM and exposure to oxygen is consistent with the antimicrobial properties of this PLA. PLA is produced by lactic acid bacteria from phenylpyruvate with phenylalanine as a precursor [34]. Our metabolomic analysis also suggested that L-phenylalanine in the TJ medium was decreased by *P. copri* JCM 13464[T] culture, whereas L-(-)-3-PLA was increased in the culture supernatant and within the bacteria. These results suggested that *P. copri* JCM 13464[T] metabolizes L-phenylalanine to produce L-(-)-3-PLA. It has been reported that *P. copri* possesses a gene cluster for enzymes involved in the production of D- and L-PLA, which has also been detected in the culture supernatant of *P. copri* [35]. Thus, cell death may have been induced in *P. copri* JCM 13464[T] cultured in TJ medium because of increased oxidative stress (from exposure to oxygen) and the presence of L-(-)-3-PLA.

Therefore, we added L-(-)-3-PLA at the same concentration (5.5 µM) detected in the TJ culture supernatant of *P. copri* JCM 13464[T] to BG medium and cultured the bacterial strains. After applying them to NGM, we evaluated their post-application viability. The addition of L-(-)-3-PLA to BG medium inhibited the growth of *S. salivarius* JCM 5707[T] after NGM application but had no observable impact on the growth or survival of *P. copri* JCM 13464[T] and *M. funiformis* JCM 14723[T]. In other words, L-(-)-3-PLA alone did not exhibit the same modulatory as TJ medium. Although the resistance mechanism of *Prevotella* to L-(-)-3-PLA has not been elucidated, the unchanged survival of *P. copri* JCM 13464[T] which produces PLA, under L-(-)-3-PLA supplementation, and the inhibited growth of *S. salivarius* JCM 5707[T], which does not produce PLA (as described in the section below), suggest that *Prevotella* may possess a specific resistance mechanism against L-(-)-3-PLA. It has been reported that the activity of superoxide dismutase increases in *Staphylococcus aureus* cultured in the presence of PLA [32]. Additionally, when the antioxidant activity gene *soxS* in *E. coli* is activated by genetic modification, the growth of *E. coli* in PLA-supplemented medium is significantly enhanced [36]. Therefore, based on these previous reports and as noted above, one of the mechanisms of the antibacterial action of PLA involves the induction of oxidative stress, and enhancing antioxidant activity could serve as a mechanism of resistance to PLA. In this study, LC-MS/MS data showed a decrease in the levels of 2,3,4,9-Tetrahydro-1H-β-carboline-3-carboxylic acid and Cyclo(phenylalanyl-prolyl) in *P. copri* JCM 13464[T] cells cultured in TJ medium, and these components have been reported to possess free radical-scavenging ability [37,38]. Therefore, it is plausible that *P. copri* JCM 13464[T] may utilize these antioxidant components as part of its resistance mechanism to L-(-)-3-PLA. Additionally, culturing in TJ medium might have led to an increase in L-(-)-3-PLA production and a reduction in antioxidant compound levels, which could have contributed to enhanced susceptibility to L-(-)-3-PLA.

It has been reported that PLA intake in mice suppresses intestinal permeability [39]. Additionally, studies on human intestinal epithelial cell lines have shown that D-PLA directly exerts an inhibitory effect on intestinal permeability [40]. In *C. elegans*, although the effect of L-(-)-3-PLA on intestinal permeability has not been reported, *C. elegans* fed 3-PLA exhibited lifespan extension, which is mediated by the activation of SKN-1/AFTS-1 [16]. Since the activation of SKN-1 in *C. elegans* has been suggested to contribute to the maintenance of intestinal barrier function [41], it was hypothesized that

L-(-)-3-PLA intake may directly suppress intestinal permeability in *C. elegans*. Therefore, in this study, *C. elegans* was fed NGM supplemented with L-(-)-3-PLA to evaluate its effect on intestinal permeability. The results showed that in the group fed 10 nM L-(-)-3-PLA, which was at the same concentration as the bacterial sample of *P. copri* JCM 13464[T] cultured in TJ medium, intestinal permeability was significantly suppressed compared to the untreated group. Thus, the mechanism by which intestinal permeability was suppressed in *C. elegans* fed *P. copri* JCM 13464[T] cultured in TJ medium may involve the enhanced bacterial lysis of *P. copri* JCM 13464[T], and the direct effect of L-(-)-3-PLA produced by *P. copri* JCM 13464[T] on enhancing intestinal barrier function.

In the present study, among *P. copri* JCM 13464[T], *M. funiformis* JCM 14723[T], and *S. salivarius* JCM 5707[T], only *P. copri* JCM 13464[T] was affected by TJ culture. This study was unable to determine the reason for the lack of effect of TJ on *M. funiformis* and *S. salivarius*. If the hypothesis regarding the mechanism of action of TJ on *P. copri* JCM 13464[T] is correct, it is possible that *Megamonas* does not produce L-(-)-3-PLA, as no reports have been published on its ability to produce L-(-)-3-PLA. There is also a report on *Streptococcus*, which states that it does not produce PLA [42]. Therefore, it is possible that the antimicrobial effect of the TJ medium culture was not exerted on these bacteria.

As for the mechanism of action of TJ, it is possible that TJ components may directly inhibit increased intestinal permeability of *C. elegans*. In practice, in this study, NGM was used for *C. elegans* rearing after a drop of concentrated *P. copri* JCM 13464[T] bacterial solution was added to the NGM and stored for 4–6 days. During this time, the liquid components in the medium and culture supernatant were expected to be absorbed into the agar and diffused into the agar. In a review of the methodology of toxicity testing using *C. elegans*, it was stated that the amount of exposure of *C. elegans* to liquid components applied to the agar medium is difficult to control because it is affected by various factors, such as humidity, solubility of the test component, and interaction with agar [43]. In this study, *C. elegans* may have been exposed to TJ components, but the amount of exposure was expected to be small and highly variable among experiments. Therefore, the results of this study may reflect differences in the conditions of the bacteria, rather than the effects of TJ components.

Finally, we discuss whether the results of this study can be replicated in humans. First, we discuss the suitability of this model for evaluating intestinal permeability. In humans, a higher prevalence of mucin-degrading bacteria in the intestine has been suggested to contribute to the disruption of intestinal barrier function [44]. *P. copri* is reported to possess genes encoding mucin-degrading enzymes, including GH20 (β-hexosaminidase), which cleaves N-acetylglucosamine/N-acetylgalactosamine [45]. Therefore, it is hypothesized that one of the mechanisms by which *P. copri* induces intestinal permeability involves the disruption of the intestinal barrier through mucin-degrading enzymes. The expression of mucin-type N-acetylgalactosamine transferases in the intestinal tract of *C. elegans* has been reported [46], and mucin-like substances composed of N-acetylgalactosamine have been detected in the *C. elegans* extracts [47], suggesting the presence of N-acetylgalactosamine-containing mucin-like substances in the *C. elegans* intestine. Therefore, it is considered that the mucin-like substances in *C. elegans* may serve as substrates for GH20, which is present in *P. copri*. From the perspective of increased intestinal permeability mediated by the degradation of intestinal mucin by *P, copri*, *C. elegans* may be considered a suitable organism that mimics human intestinal function. In fact, *Pseudomonas aeruginosa*, which possesses a potent array of mucin-degrading enzymes [48], has been implicated in the disruption of the intestinal barrier in humans, leading to infection [49]. Moreover, *Pseudomonas aeruginosa* has been reported to degrade mucin [50] and increase intestinal permeability in *C. elegans* [51]. Next, we discuss whether the effects of culturing *P. copri* in TJ medium observed in this study can be replicated in the human intestine. As mentioned above, the combination of L-(-)-3-PLA and oxidative stress has been suggested to induce bacterial lysis in *P. copri* Oxygen levels in the intestinal tract are extremely low [52], and there is concern that the oxidative stress experienced by intestinal bacteria is minimal. However, it has been suggested that in mice, intestinal D-amino acid oxidase metabolizes D-amino acids derived from intestinal bacteria to generate hydrogen peroxide, which in turn reduces the abundance of Bacteroidales (a superclass of *Prevotella*) in the intestinal tract [53]. Thus, it is possible that intestinal bacteria are affected by oxidative stress in the intestinal tract, and the decrease in *Prevotella* observed in this study may also be replicated in the intestinal tract. As mentioned above, based on

the appropriateness of the evaluation model and the proposed mechanism of action of TJ on *P. copri*, it is hoped that the phenomena observed in this study may ultimately be replicated in humans. However, this study was conducted using *C. elegans* as a model organism and single bacterial strains *in vitro*, without considering host digestive processes and bacterial interactions in the gut, so extrapolation of these findings to humans remains a challenge for future research.

In summary, the results of this study suggest that TJ may inhibit intestinal permeability by both rendering *P. copri* JCM 13464$^T$ vulnerable and inducing the production of L-(-)-3-PLA, which has intestinal barrier-protective effects, by *P. copri* JCM 13464$^T$In future human intervention studies, it is necessary to confirm whether tomato consumption leads to a reduction in the abundance of *Prevotella* in the intestinal microbiota and an improvement in the intestinal permeability. The mechanism by which tomatoes induce *Prevotella* vulnerability needs to be elucidated in more detailed experiments under physiological conditions.

## Supporting information

**S1 Table. Body size of *Caenorhabditis elegans* on days 4–8 of adulthood, after exposure to test bacteria from hatching.** *Caenorhabditis elegans* were fed test bacterial strains cultured on either BG or TJ medium from the time of hatching until day 8 post-hatching. Body size was measured from day 4 to day 8 post-hatching.
(XLSX)

**S2 Table. Brood size of *Caenorhabditis elegans* on days 4–8 of adulthood, after exposure to test bacteria from hatching.** *Caenorhabditis elegans* were fed test bacterial strains cultured on either BG or TJ medium from the time of hatching until day 8 post-hatching. Brood size was measured from day 4 to day 8 post-hatching.
(XLSX)

**S3 Table. Flow cytometric analysis of *Prevotella copri* JCM 13464$^T$ washed with sterile distilled water.** Numerical data of the means and standard deviations were used to create the graphs in Fig 4.
(XLSX)

**S4 Table. Effect of culturing in a BG medium supplemented with tomato juice or L-(-)-3-phenyllactic acid on the growth or survival rate of the tested bacterial strains.** Numerical data of the means and standard deviations were used to create the graphs in Fig 6.
(XLSX)

**S5 Table. The results of Metabolomic analysis of *Prevotella copri* JCM 13464$^T$.** All components with a mzCloud best-match confidence of more than 70% from the LC-MS/MS analysis of the medium itself, culture supernatant, or cells of *Prevotella copri* JCM 13464$^T$ were listed.
(XLSX)

## Acknowledgments

We express our sincere gratitude to Dr. Daichi Kokubu and Dr. Takuro Inoue for their careful review and valuable comments on our manuscript.

## Author contributions

**Conceptualization:** Nobuo Fuke, Natsumi Desaka, Yasuki Higashimura.

**Formal analysis:** Nobuo Fuke, Yuichiro Nakazawa, Kenji Matsumoto, Yasuki Higashimura.

**Investigation:** Nobuo Fuke, Yuichiro Nakazawa, Kenji Matsumoto, Yasuki Higashimura.

**Supervision:** Shigenori Suzuki, Yasuki Higashimura.

**Writing – original draft:** Nobuo Fuke.

**Writing – review & editing:** Natsumi Desaka, Yuichiro Nakazawa, Shigenori Suzuki, Kenji Matsumoto, Yasuki Higashimura.

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
