## [Decision Letter · Decision Letter 0]

19 Jan 2025

PONE-D-24-56842

Tomato juice supplemented cultivation of *Prevotella copri* inhibits intestinal permeability enhancing effect of the bacteria in *Caenorhabditis elegans*

PLOS ONE

Dear Dr. Fuke,

Thank you for submitting your manuscript to PLOS ONE. After careful consideration, we feel that it has merit but does not fully meet PLOS ONE’s publication criteria as it currently stands. Therefore, we invite you to submit a revised version of the manuscript that addresses the points raised during the review process.

Both reviewers' have provided constructive comments to improve the manuscript. Specifically, more explanation should be provided for the study design, including the use of 1% v/v tomato juice. Reviewer #1 further pointed out the need to examine the viable counts of the three bacterial species. In addition, reviewer #2 commented the lack of clarity for the statistical analyses used. 

We look forward to receiving your revised manuscript.

Kind regards,

Chun Wie Chong

Academic Editor

PLOS ONE

Reviewers' comments:

Reviewer's Responses to Questions

**Comments to the Author**

1. Is the manuscript technically sound, and do the data support the conclusions?

Reviewer #1: Partly

Reviewer #2: Partly

2. Has the statistical analysis been performed appropriately and rigorously? 

Reviewer #1: Yes

Reviewer #2: Yes

3. Have the authors made all data underlying the findings in their manuscript fully available?

Reviewer #1: Yes

Reviewer #2: Yes

4. Is the manuscript presented in an intelligible fashion and written in standard English?

Reviewer #1: No

Reviewer #2: No

5. Review Comments to the Author

Reviewer #1: In a previous epidemiological study, the authors investigated the relationship between human intestinal bacteria, tomato juice intake, and blood LPS-binding protein concentrations. As a result, they reported that there was a positive correlation between the population of intestinal bacteria three genera, Prevotella, Megamonas, and Streptococcus, and LPB, and a negative correlation between the intake of tomato juice and the amount of bacteria genus Streptococcus. Therefore, the authors used a model organism, the nematode C. elegans, to feed these bacteria cultured in a medium containing tomato juice and examined the intestinal permeability. The ideas in this study are considered novel and important. On the other hand, some controlled experiments and more careful consideration are needed to reach any conclusions. Additionally, the English text also needs further proofreading.

(Major points)

(1) The title and short title are unclear and should be corrected.

For example, “Cultivation of P. copri in a medium supplemented with tomato juice suppresses the bacteria-induced intestinal permeability in C. elegans.” or “P. copri cultured with tomato juice supplement abolishes bacteria-induced intestinal permeability in C. elegans.”

(2) To assert that the three bacteria used in this experiment increase the intestinal permeability of C. elegans, the results of the Smurf assay of E. coli OP-50 cultured for up to 8 days as a control are needed.

(3) In addition, the authors should conduct experiments in which these three types of bacteria are fed to L1 larval nematodes from day 1 to day 8. Whether the nematodes can digest these bacteria and grow normally is also important for understanding the physiological state of the nematodes in the experiment in which bacteria are added from day 4.

(4) Could the large indentations seen in SEM observations (Fig. 3) be debris of bacterial origin? Why would washing bacteria with sterile water produce indentations? An explanation is needed.

(5) Furthermore, shouldn't saline be used in general to wash away bacteria? Osmotic pressure changes occur. Washing PI-positive cells with sterile water may cause the bacteria to burst and die due to osmotic stress, leading to a decrease in PI positivity and an increase in debris.

(6) The decrease in absorbance seen in Figure 6 may be due to the lysis of bacteria during storage. Furthermore, this lysis appears to be accelerated by culturing with TJ. In the washing with sterile water (Figures 3 and 4 exp) and the experiment in Figure 6, it is necessary to measure the number of live bacteria by measuring the number of colonies formed.

(7) One hypothesis is that when cultured with the addition of tomato juice, the bacteria produces antibiotics, but this hypothesis is difficult to understand because antibiotic-producing bacteria generally have mechanisms for conferring resistance to the antibiotics themselves.

(8) It remains to be confirmed whether heat treatment of the other two bacteria also alters the intestinal permeability of the nematode. This finding may simply be the result of P.copri bacteria being more easily lysed (dead) when cultured in a medium containing tomato juice and washed with sterile water. Examining the viable counts of the three bacteria after each treatment may provide the key to solving this mystery.

Reviewer #2: In this study, the authors address a novel aspect of how tomato consumption may influence intestinal permeability through its effect on specific gut bacteria, which is relevant given the increasing interest in diet-microbiota interactions. However, the following comments and suggestions might be useful for improving the manuscript.

Major comments:

1. Utilizing C. elegans as a model organism for studying the impact of bacterial composition on intestinal permeability provides an efficient system for preliminary screening before moving to more complex models. However�the authors note that extrapolation to humans remains a challenge, it would be beneficial to discuss potential pathways or mechanisms that could be conserved across species, thereby justifying further studies in more complex models or human populations.

2. Although L-(-)-3-phenyllactic acid accumulation is identified as a possible factor, more detailed investigation into how this compound specifically affects intestinal permeability and whether it directly causes increased vulnerability of P. copri would enhance the study's explanatory power.

3. Providing information about the statistical methods used to analyze the differences in Smurf ratios and survival rates would increase confidence in the reported outcomes.

6. PLOS authors have the option to publish the peer review history of their article (what does this mean? ). If published, this will include your full peer review and any attached files.

**Do you want your identity to be public for this peer review?** For information about this choice, including consent withdrawal, please see our Privacy Policy .

Reviewer #1: **Yes: ** Atsushi Higashitani

Reviewer #2: No

---

## [Author Response · Author response to Decision Letter 1]

15 Jul 2025

Response to Reviewers

We thank the academic editor and reviewers for the time and effort spent on the manuscript. The suggestions and comments have improved the quality of the manuscript. We have made responses to the comments point-by-point, and revised the manuscript accordingly, as follows. We hope our revisions fully meet your requirements.

Response to the academic editor

Point 1: Specifically, more explanation should be provided for the study design, including the use of 1% v/v tomato juice.

Response: Thank you very much for your comment. We chose to use commercially available tomato juice as the test sample because it provides more consistent quality compared to fresh tomato juice, which enhances reproducibility in future human intervention studies. Additionally, based on our previous research, we have found that supplementing the gut microbiota culture system with food ingredients at a concentration of 1% (v/v) is sufficient to induce changes in the composition of various bacterial species. At this concentration, the effect of the test ingredient can be evaluated without significantly disturbing the nutrient balance of the BG medium. For these reasons, we used tomato juice at a concentration of 1% (v/v) in this study. We have added this explanation to the revised manuscript.

Details of corrections in “traced” manuscript:

(Page 6, Lines 91-97 in the revised manuscript with track changes)

The sentences have been added as follows: “Commercially available tomato juice was selected as the test sample due to its consistent quality compared to fresh juice and its suitability for use in future human intervention studies. Additionally, based on our previous findings, we confirmed that adding food ingredients to the gut microbiota culture system at a concentration of 1% (v/v) is sufficient to alter the composition of various bacterial species. Furthermore, at a concentration of 1% (v/v), we believe the impact on bacteria can be assessed without disrupting the nutrient composition inherent to the BG medium. Thus, a 1% (v/v) concentration of tomato juice was chosen for this study.”

Point 2: Reviewer #1 further pointed out the need to examine the viable counts of the three bacterial species.

Response: Thank you very much for your comment. In response to Reviewer #1’s suggestion, we attempted a colony formation assay for P. copri, the primary focus of this study. However, no colonies were observed. Given that P. copri is an obligate anaerobe, we suspect that its exposure to atmospheric oxygen after application to the NGM plates compromised its ability to form colonies. Therefore, in this study, we assessed bacterial counts by measuring turbidity. Further details can be found in our response to Reviewer #1, Point 7.

Point 3: In addition, reviewer #2 commented the lack of clarity for the statistical analyses used.

Response: Thank you very much for your comment. As mentioned in our response to Reviewer #2, Point 4, the statistical methods used for each analysis are described in the Materials and Methods section and in the respective figure legends. However, we agree that the rationale for choosing specific statistical tests could be more clearly explained. In particular, when comparing continuous variables between groups, it would have been more appropriate to first assess the equality of variances before selecting the statistical method.

Therefore, we have revised the statistical analysis procedures as follows: For comparisons between two groups, an F-test was performed to assess the equality of variances. For comparisons among three groups, Bartlett’s test for homogeneity of variances was used. If a tendency toward unequal variances was observed (P < 0.1), Welch’s t-test was applied for both two- and three-group comparisons. In the case of three-group comparisons, P-values were adjusted using the Holm method. If no significant difference in variances was detected (P ≥ 0.1), indicating that the assumption of equal variances was met, Student’s t-test was used for two-group comparisons, and Dunnett’s test was used for three-group comparisons. We have updated the Materials and Methods section accordingly in the revised manuscript. Further details can be found in our response to Reviewer #2, Point 4.

Response to Reviewer #1

Point 1: In a previous epidemiological study, the authors investigated the relationship between human intestinal bacteria, tomato juice intake, and blood LPS-binding protein concentrations. As a result, they reported that there was a positive correlation between the population of intestinal bacteria three genera, Prevotella, Megamonas, and Streptococcus, and LPB, and a negative correlation between the intake of tomato juice and the amount of bacteria genus Streptococcus. Therefore, the authors used a model organism, the nematode C. elegans, to feed these bacteria cultured in a medium containing tomato juice and examined the intestinal permeability. The ideas in this study are considered novel and important. On the other hand, some controlled experiments and more careful consideration are needed to reach any conclusions. Additionally, the English text also needs further proofreading.

Response: Thank you very much for your valuable comments and constructive feedback. We have now conducted comprehensive English language proofreading of the entire manuscript to improve clarity and readability, and all corrections made during this process are retained as track changes in the Word file.

Point 2: The title and short title are unclear and should be corrected.

For example, “Cultivation of P. copri in a medium supplemented with tomato juice suppresses the bacteria-induced intestinal permeability in C. elegans.” or “P. copri cultured with tomato juice supplement abolishes bacteria-induced intestinal permeability in C. elegans.”

Response: Thank you very much for your comment. In accordance with your recommendation, we have revised both the manuscript title and the short title to improve clarity and better reflect the content of the study.

Details of corrections in “traced” manuscript:

(Page 1, Lines 1-4 in the revised manuscript with track changes)

The sentences have been corrected as follows: Tomato juice supplemented cultivation of Prevotella copri inhibits intestinal permeability enhancing effect of the bacteria in Caenorhabditis elegansCultivation of Prevotella copri in a medium supplemented with tomato juice suppresses the bacteria-induced intestinal permeability in Caenorhabditis elegans

(Page 1, Lines 6-7 in the revised manuscript with track changes)

The sentences have been corrected as follows: Tomato juice cultivation inhibits P. copri's effect on nematode intestinal permeabilityTomato juice-supplemented culture reduces P. copri-induced intestinal permeability in C. elegans

Point 3: To assert that the three bacteria used in this experiment increase the intestinal permeability of C. elegans, the results of the Smurf assay of E. coli OP-50 cultured for up to 8 days as a control are needed.

Response: Thank you very much for your comment. In response, we conducted the Smurf assay using E. coli OP50 and compared the proportion of Smurf individuals to those observed in worms fed with Prevotella, Megamonas, and Streptococcus. As a result, no significant differences were observed in the proportion of Smurf individuals between the E. coli OP50 group and any of the other bacterial groups.

E. coli OP50 is widely used as the standard laboratory diet for C. elegans due to its slow growth and the formation of a thin bacterial lawn on NGM agar, which facilitates microscopic observation. It is also favored for its safety and ease of handling. However, wild C. elegans are naturally associated with a diverse gut microbiota. Previous studies have shown that feeding worms a bacterial mixture resembling their natural microbiota can extend lifespan compared to worms fed with E. coli OP50 (Stover MA et al., Front Physiol. 2023 Sep 12;14:1207705). These findings suggest that E. coli OP50 may not be optimal for maintaining gut barrier function.

Therefore, in the present study, we did not place primary emphasis on comparisons with E. coli OP50. Instead, we focused on assessing the effects of culturing each bacterial strain in TJ medium and their individual influence on host intestinal permeability. We have added these experimental results and their interpretation to the revised manuscript.

Details of corrections in “traced” manuscript:

(Page 2, Lines 23-27 in the revised manuscript with track changes)

The underlined sentence has been added as follows: We cultured Escherichia coli OP50 (the standard C. elegans food), Prevotella (P.) copri JCM 13464T, Megamonas funiformis JCM 14723T, and Streptococcus salivarius JCM 5707T in either normal medium or medium containing 1% (v/v) tomato juice (TJ medium), fed these bacteria to C. elegans for three days, and evaluated intestinal permeability using the Smurf assay.

(Page 7, Lines 115-117 in the revised manuscript with track changes)

The underlined sentence has been added as follows: The bacterial strains used in this study are summarized in Table 1. All strains, except for E. coli OP50, were obtained from the Microbe Division/Japan Collection of Microorganisms (JCM), RIKEN BioResource Research Center (Ibaraki, Japan).

(Page 22, Lines 369-378 in the revised manuscript with track changes)

The underlined sentence has been added as follows: E. coli OP50, P. copri JCM 13464T, M. funiformis JCM 14723T, and S. salivarius JCM 5707T were grown anaerobically in BG or TJ medium, concentrated, and added to NGM. C. elegans were fed E. coli OP50, the standard diet, for 4 days after hatching, and then fed the test organisms applied to NGM from days 5 to 8. On day 8, intestinal permeability of C. elegans was evaluated using the Smurf assay (Fig 1A). For P. copri JCM 13464T, the proportion of Smurfs was significantly lower in C. elegans that had ingested bacteria grown on the TJ medium than in those grown on the BG medium (Fig 1B). For E. coli, M. funiformis and S. salivarius, no significant differences were observed between the two media. No significant difference in the proportion of Smurf individuals was observed between E. coli OP50 and P. copri, M. funiformis, or S. salivarius cultured on BG medium.

(Page 35, Lines 600-614 in the revised manuscript with track changes)

The sentences have been added as follows: As noted in the Introduction, Prevotella, Megamonas, and Streptococcus are gut bacterial genera implicated in increased intestinal permeability in human epidemiological studies. Therefore, we compared the intestinal permeability of C. elegans fed these bacteria with that of worms fed E. coli OP50, the standard laboratory diet for C. elegans. However, no significant differences were observed in the proportion of "Smurf" phenotype individuals between worms fed E. coli OP50 and those fed P. copri JCM 13464T, M. funiformis JCM 14723T, or S. salivarius JCM 5707T. E. coli OP50 is commonly used as the standard food source for C. elegans because it grows slowly and forms a thin bacterial lawn on NGM agar plates, facilitating microscopic observation of the worms. Moreover, it is well-suited for laboratory use due to its safety and ease of cultivation. Nonetheless, wild C. elegans harbor diverse gut microbiota. Previous studies have shown that feeding C. elegans a bacterial mixture resembling the natural gut microbiota of wild nematodes can extend lifespan compared to worms fed E. coli OP50 [28]. This suggests that E. coli OP50 may not be the optimal diet for maintaining gut barrier function. Therefore, this study does not focus on direct comparisons between E. coli OP50 and the other tested bacterial strains. Instead, we focused on the effects of culturing each bacterial species in TJ medium, emphasizing their individual impact on host intestinal permeability.

(Figures)

The results of the Smurf assay conducted on C. elegans fed with E. coli OP50 cultured in either BG or TJ medium have been added to Fig. 1B.

Point 4: In addition, the authors should conduct experiments in which these three types of bacteria are fed to L1 larval nematodes from day 1 to day 8. Whether the nematodes can digest these bacteria and grow normally is also important for understanding the physiological state of the nematodes in the experiment in which bacteria are added from day 4.

Response: Thank you very much for your comment. To assess the potential effects of the test bacterial strains on C. elegans development, we conducted experiments in which the strains were administered from the time of hatching (L1 stage). In control animals fed E. coli OP50 cultured on BG medium, body size increased steadily through day 5 post-hatching (S1 Table). In contrast, worms fed with Prevotella, Megamonas, or Streptococcus from hatching exhibited slower somatic growth. Moreover, Prevotella-fed worms showed a pronounced reduction in egg production (S2 Table).

Based on these findings, to minimize potential confounding effects of the test strains on worm development, we chose to begin bacterial administration on day 5 post-hatching, when somatic growth is largely complete. This experimental rationale and the relevant data have been added to the revised manuscript.

Details of corrections in “traced” manuscript:

(Pages 10-11, Lines 165-173 in the revised manuscript with track changes)

The sentences have been added as follows: To evaluate the potential effects of the test bacterial strains on C. elegans development, we examined whether administering these strains from the time of hatching would influence worm growth. In control animals fed E. coli OP50 cultured on BG medium, body size increased steadily through day 5 post-hatching (S1 Table). In contrast, worms fed P. copri JCM 13464T, M. funiformis JCM 14723T, or S. salivarius JCM 5707T from hatching exhibited slower body growth. Furthermore, worms fed P. copri JCM 13464T exhibited a marked reduction in egg production (S2 Table). Based on these findings, to minimize any confounding effects of the test strains on somatic development, we began administration of the bacteria on day 5 post-hatching, when body growth is considered to plateau. The final C. elegans feeding schedule was as follows:

(Page 51, Lines 960-967 in the revised manuscript with track changes)

The sentences have been added as follows:

S1 Table. Body size of Caenorhabditis elegans on days 4–8 of adulthood, after exposure to test bacteria from hatching. Caenorhabditis elegans were fed test bacterial strains cultured on either BG or TJ medium from the time of hatching until day 8 post-hatching. Body size was measured from day 4 to day 8 post-hatching.

S2 Table. Brood size of Caenorhabditis elegans on days 4–8 of adulthood, after exposure to test bacteria from hatching. Caenorhabditis elegans were fed test bacterial strains cultured on either BG or TJ medium from the time of hatching until day 8 post-hatching. Brood size was measured from day 4 to day 8 post-hatching.

(Tables)

The evaluation results of body size and brood size in C. elegans have been included in S1 and S2 Tables, respectively.

Point 5: Could the large indentations seen in SEM observations (Fig. 3) be debris of bacterial origin? Why would washing bacteria with sterile water produce indentations? An explanation is needed.

Response: Thank you very much for your comment. It has been reported that in E. coli, exposure to a hypotonic environment leads to an influx of water into the cytoplasm, which activates mechanosensitive channels (Hartmann M. et al., Antimicrob Agents Chemother. 2010 Aug;54(8):3132–42). This activation results in the release of water and ions into the periplasmic space, causing it to expand. The subsequent structural changes in the outer membrane may appear as surface undulations or localized indentations under SEM. Similar morphological features were observed in P. copri JCM 13464T following washing with sterile distilled water, as shown in Fig. 3. These observations suggest that the indentations are not bacterial debris but rather a physiological response to osmotic stress. We have included this explanation in the revised manuscript.

Details of corrections in “traced” manuscript:

(Pages 25-26, Lines 435-4

---

## [Decision Letter · Decision Letter 1]

17 Aug 2025

Cultivation of *Prevotella copri* in a medium supplemented with tomato juice suppresses the bacteria-induced intestinal permeability in *Caenorhabditis elegans*

PONE-D-24-56842R1

Dear Dr. Fuke,

We’re pleased to inform you that your manuscript has been judged scientifically suitable for publication and will be formally accepted for publication once it meets all outstanding technical requirements.

Kind regards,

Chun Wie Chong

Academic Editor

PLOS ONE

Additional Editor Comments (optional):

Reviewers' comments:

Reviewer's Responses to Questions

**Comments to the Author**

1. If the authors have adequately addressed your comments raised in a previous round of review and you feel that this manuscript is now acceptable for publication, you may indicate that here to bypass the “Comments to the Author” section, enter your conflict of interest statement in the “Confidential to Editor” section, and submit your "Accept" recommendation.

Reviewer #1: All comments have been addressed

Reviewer #2: All comments have been addressed

2. Is the manuscript technically sound, and do the data support the conclusions?

Reviewer #1: Yes

Reviewer #2: Yes

3. Has the statistical analysis been performed appropriately and rigorously? 

Reviewer #1: Yes

Reviewer #2: Yes

4. Have the authors made all data underlying the findings in their manuscript fully available?

Reviewer #1: Yes

Reviewer #2: Yes

5. Is the manuscript presented in an intelligible fashion and written in standard English?

Reviewer #1: Yes

Reviewer #2: Yes

6. Review Comments to the Author

Reviewer #1: (No Response)

Reviewer #2: (No Response)

7. PLOS authors have the option to publish the peer review history of their article (what does this mean? ). If published, this will include your full peer review and any attached files.

**Do you want your identity to be public for this peer review?** For information about this choice, including consent withdrawal, please see our Privacy Policy .

Reviewer #1: **Yes: ** Atsushi Higashitani

Reviewer #2: **Yes: ** Qiuhong Niu

---

## [Editor Report · Acceptance letter]

PONE-D-24-56842R1

PLOS ONE

Dear Dr. Fuke,

I'm pleased to inform you that your manuscript has been deemed suitable for publication in PLOS ONE. Congratulations! Your manuscript is now being handed over to our production team.

Kind regards,

on behalf of

Dr. Chun Wie Chong

Academic Editor

PLOS ONE